# Tuning the Surface Plasmon Resonance of Lanthanum Hexaboride to Absorb Solar Heat: A Review

**DOI:** 10.3390/ma11122473

**Published:** 2018-12-05

**Authors:** Tracy M. Mattox, Jeffrey J. Urban

**Affiliations:** Molecular Foundry, Lawrence Berkeley National Laboratory, Berkeley, CA 94720, USA

**Keywords:** lanthanum hexaboride, LaB_6_, plasmon, nanoparticles, heat absorption

## Abstract

While traditional noble metal (Ag, Au, and Cu) nanoparticles are well known for their plasmonic properties, they typically only absorb in the ultraviolet and visible regions. The study of metal hexaborides, lanthanum hexaboride (LaB_6_) in particular, expands the available absorbance range of these metals well into the near-infrared. As a result, LaB_6_ has become a material of interest for its energy and heat absorption properties, most notably to those trying to absorb solar heat. Given the growing popularity of LaB_6_, this review focuses on the advances made in the past decade with respect to controlling the plasmonic properties of LaB_6_ nanoparticles. This review discusses the fundamental structure of LaB_6_ and explains how decreasing the nanoparticle size changes the atomic vibrations on the surface and thus the plasmonic absorbance band. We explain how doping LaB_6_ nanoparticles with lanthanide metals (Y, Sm, and Eu) red-shifts the absorbance band and describe research focusing on the correlation between size dependent and morphological effects on the surface plasmon resonance. This work also describes successes that have been made in dispersing LaB_6_ nanoparticles for various optical applications, highlighting the most difficult challenges encountered in this field of study.

## 1. Introduction

Traditional plasmonic metals (Ag, Au, and Cu) possess enormous free carrier densities, and when confined on the nanoscale the quantized free electron oscillations result in sharp localized surface plasmon resonance (LSPR) modes. The LSPR properties enhance light-matter interactions, making these materials ideal for a wide variety of electronic and optical applications [1,2,3,4]. Furthermore, their sensitivity to small changes within their structure (e.g., size, morphology, atomic vacancies, etc.) makes the properties easy to tune by introducing defects or changing the surface by varying the size or shape.

The ability of plasmonic metals to convert solar light into electricity and chemical energy is well documented, but the absorbance is restricted to the ultraviolet and visible spectrums in traditional plasmonic metals (Figure 1). This leaves the near infrared (NIR) region mostly inaccessible to metal nanoparticles, with the exception of novel engineered geometries of some metals, such as the cases of Au nanowires and shells [5,6]. There is a growing need to find materials with a NIR absorbance for applications such as window coatings to absorb solar heat [7,8]. Researchers attempting to reduce heat entering buildings and automobiles through windows need a visibly transparent material that absorbs the most intense radiative heat from the sun, ideally in the range of 750–1200 nm. While some chalcogenides and metal oxides are potential candidates [9,10,11,12,13,14], the metal borides are often overlooked.

Lanthanum hexaboride (LaB_6_) is a plasmonic metal with a large free carrier density [20,21,22] that is best known for its impressive thermionic emission properties and low work function [23,24,25,26,27,28,29]. However, LaB_6_ also absorbs light very strongly at about 1000 nm, which falls well within the targeted range to absorb solar heat [30,31]. The optical properties of LaB_6_ nanoparticles coupled with its incredible hardness [32,33,34,35,36] and high thermal stability [37,38,39,40,41,42,43] make it an excellent choice to include in alloys and composites for solar window applications.

The ability to directly synthesize LaB_6_ on the nanoscale has only recently become a reality, so research focusing on plasmonic control in LaB_6_ is relatively new. That said, LaB_6_ does offer the same wide range of methods for optical tuning as other LSPR particles, including controlling the carrier density through vacancies and doping, particle size, morphology, and the surrounding media (i.e. ligands and polymer matrices) [10,44,45,46,47]. There is still much to be learned about the optical properties of LaB_6_ nanoparticles, but what has been discovered thus far has been quite exciting and has made LaB_6_ a very popular material of interest in recent years. 

This review focuses on the advances that have been made in the last decade with respect to tuning the plasmon of LaB_6_ nanoparticles. Though this field is still relatively new, we feel that the growing demand for optically tunable LaB_6_ warrants a comprehensive review to explain the nuances of controlling the plasmon of LaB_6_. This work describes how some of the fundamentals behind this intriguing material correspond with the optical properties, and how doping, size, and morphology are contributing factors when attempting to meet requirements of desired plasmonic applications.

## 2. Relating LaB_6_ Fundamentals to Plasmonics

### 2.1. Crystal Structure of LaB_6_

LaB_6_ is composed of interconnecting hexaboride clusters with lanthanum (La) atoms residing in the interstitial spaces (Figure 2A) [48]. It is well known that the boron network is responsible for the rigidity of the structure, and consequently the thermal and mechanical stability. Though La is not bound within the system it does provide electrons to stabilize the structure. Interestingly, this rigid material is more relaxed on the particle surface (Figure 2B) [49,50,51]. On the outermost surface, lanthanum atoms relax closer to the B_6_ framework while the B_6_ octahedral clusters relax and expand slightly outwards. Though this relaxation is insignificant to the optical properties of bulk sized LaB_6_, the surface vibrations of nanoparticles are incredibly important when attempting to control the LSPR.

Researchers are still exploring the surface chemistry of LaB_6_, and remain uncertain of the mechanistic details of crystal formation and growth. Until recently, it was assumed that high temperatures (>1500 °C) were required to make LaB_6_, most often by reacting lanthanum salt (e.g., LaCl_3_) with sodium borohydride (NaBH_4_). Unfortunately, the quick nucleation and growth at such high temperatures prevents the direct synthesis of nano-sized LaB_6_. In typical nanoparticle syntheses, it is important to control the rate of nucleation and growth in order to make particles of a uniform size. This is often accomplished by varying temperature, concentration, or including additives (ligands) [52,53]. Unfortunately, the assumed high temperature requirements make small adjustments traditionally used in nanoparticle work impossible, especially the incorporation of organic ligands that cannot withstand the extreme heat. As a result, researchers have had to rely on ball milling techniques to reduce the size of LaB_6_, which introduces contaminants by the very nature of the process [54,55,56]. In recent years, researchers have found that significantly lowering the heat can yield phase-pure LaB_6_ nanoparticles using a variety of methods, including tube furnaces [57,58], autoclaves [59,60], and vapor deposition [61,62]. In reducing the reaction temperature, the nucleation and growth rate is significantly reduced, making it possible to observe crystal lattice formation through in-situ diffraction measurements [48]. Researchers have recently discovered that in low temperature reactions the halogen on the lanthanum salt acts as a bridging ligand between La atoms [48,58,63]. When these LaB_6_ nanoparticles are heated, the halogen atoms force the lattice to contract until the halogen is removed (Figure 2C). This has important implications to LSPR studies, which are incredibly sensitive to the particle surface.

#### 2.1.1. Vibrational Structure of LaB_6_

Decreasing the particle size of LaB_6_ increases the surface area, and in the case of extremely small sizes it becomes possible for the nanoparticles to be composed almost entirely of surface atoms. As a result, minor changes to the surface composition can have a significant impact on the lattice and consequently the LSPR. In order to use this concept to fine tune the position of the plasmon, it is essential to first describe the vibrational modes of LaB_6_, which are displayed in Equation (1).
Γ = A_1g_ + E_g_ + T_1g_ + T_2g_ + 3T_1u_ + T_2u_(1)

As seen in Figure 3A, the vibrational modes of the B_6_ cluster in LaB_6_ include the bending (T_2g_) and stretching (E_g_ and A_1g_) modes, while La includes vibrations from moving within the boron cage (“rattling mode”) and movement with respect to the boron cage (T_1u_). When LaB_6_ particles decrease in size from bulk to 2.5 nm, there is a shift to higher energy for all of the B_6_ cluster vibrational modes due to the increased surface area which has a larger number of relaxed B_6_ on the surface (Figure 3B). Furthermore, different sized halogens from the La salt precursor also impact the vibration modes, where larger atoms take up more space as bridging atoms that increase the vibrational energy. More details on the direct influence of particle size on the plasmon position will be discussed below.

#### 2.1.2. Structural Defects

LaB_6_ is well known to have defects within the structure, and is often non-stoichiometric because La atoms are missing or excess boron is residing in the interstitial spaces. As with other plasmonic systems, these defects can be advantageous to scientists when tuning the optical properties so long as the defects can be predicted and controlled. Only recently have researchers published on the ability to control La defects simply through reaction temperature and heating rate [64]. When the system is not stoichiometric (6B:1La) and the %B is increased (La atoms are missing) the Raman vibrations shift to lower energy. This is very clear in Figure 3C when comparing 6.2 nm particles containing 38.0% and 33.5% B (note that stoichiometric LaB_6_ contains 31.8% B). The vacant lanthanum positions of the structure lower the overall energy of the system, and with less La available there are fewer electrons contributing to the structure, which causes the plasmon to shift.

## 3. Controlling the Plasmon of LaB_6_

The structure of LaB_6_ is exceptionally robust. Although vacancies and slight changes to the crystal structure clearly influence the position of the plasmon, it is possible to move beyond the fundamental structure to find other knobs to control the plasmonic properties. Similar to chalcogenides and metal oxides, LaB_6_ is significantly influenced by the particle size and the shape of the surface. Unlike traditional noble-metal based materials, LaB_6_ has the ability to incorporate dopants within its structure, which offers an additional means to increase the free carrier density and red-shift the frequency of the plasmon.

### 3.1. Doping

LaB_6_ is well known for its stability, even when La vacancies leave unfilled holes within the crystal lattice. Like all boron frameworks, the B_6_ network is electron-deficient and stable only as a result of electron transfer from the surrounding metals [65]. The natural defects or holes found in the LaB_6_ framework make this material ideal for incorporating other metals via doping to tune the optical properties. However, few people have been successful in producing phase-pure hexaboride nanoparticles containing more than one type of metal, and fewer still have considered plasmonic changes in these doped systems. Fortunately, the introduction of dopants into LaB_6_ offers a promising route to plasmon control.

Part of the difficulty arises from the nature of the synthesis. The mechanism of formation is not understood, and it is only in very recent years that researchers have found ways to control nucleation and growth in LaB_6_ by significantly reducing the reaction temperature. The other difficulty in doping LaB_6_ is that the metal does not form a direct bond with the rigid B_6_ framework. As a result, LaB_6_ is not influenced much by differing sizes of metal atoms that could alter the structure and shift the plasmon. The contribution of electrons from the metal to the structure also means that attempting to dope the trivalent La^3+^ containing LaB_6_ with other trivalent metals will not shift the plasmon because the free electron density will not change [66]. Consequently, researchers must be very selective when designing doped LaB_6_ nanoparticles when intending to tune the plasmon. Doping this system also changes the vacancies within the crystal lattice, so careful attention must be paid to determine whether the dopants or vacancies are responsible for changes to the optical properties.

Studies of the Fermi level may help provide insight into the movement of electrons within the system. The spatial distribution of electrons near the Fermi level was recently reported for trivalent LaB_6_ and divalent BaB_6_ [67], and weak electron lobes were found around the interior-B_6_ octahedral bond (Figure 4A). Comparing theoretical and experimental work, it was determined that these electron lobes were responsible for the conductive π-electrons in LaB_6_. Trivalent LaB_6_ behaves as a metal while divalent hexaborides (i.e. BaB_6_) are semiconductors [68], but how do these and the optical properties change when LaB_6_ becomes a mixed valence system? Below are the few examples in the literature that tune the plasmon of LaB_6_ nanoparticles by doping the material with divalent lanthanide metals, including some comments on how the changed Fermi surface influences the LSPR. In all cases, introducing a lanthanide metal as a dopant in the system causes a red-shift of the plasmon.

#### 3.1.1. Yb-Doped LaB_6_ Nanoparticles: Theory

A recent report used density functional theory (DFT) first-principal calculations to study the potential optical effects of LaB_6_ nanoparticles doped with ytterbium (Yb), and found that the 4f states of the Yb dopant at the Fermi surface have the potential to uniquely influence the optical properties, due to the participation of f-orbitals [69]. This is clear in Figure 4B, which compares phase pure YbB_6_, LaB_6_, and La_0.625_Yb_0.375_B_6_ in the energy loss spectra (another method for measuring plasmon energies). Doping LaB_6_ with Yb is expected to reduce the plasmon energy of the system, coinciding with a change of the LSPR. The concentration of charge carriers in the system decreases and reduces the plasma frequency when some of the La atoms are replace with Yb, which causing the minimum plasma absorption to shift to higher wavelengths. Though the calculations suggest that Yb-doping LaB_6_ should indeed tune the plasmon, such an experiment has yet to be reported.

#### 3.1.2. Sm-Doped LaB_6_ Nanoparticles 

The first synthetic example of LaB_6_ nanoparticles doped with a trivalent metal was reported using samarium (Sm) [70]. With the aid of DFT calculations, it was determined that the shift of the optical properties was due to the changing Sm 4f states near the Fermi surface of LaB_6_ after doping. The same as was predicted for Yb doping, with a reduced number of conduction electrons causing the shift of absorption spectra. When increasing the Sm content in SmB_6_ the plasmon peak shifts to higher wavelengths, which is clear as the onset of the absorbance peak shifts from 603 nm for LaB_6_ to 756 nm for SmB_6_ (Figure 4C).

#### 3.1.3. Eu-Doped LaB_6_ Nanoparticles

A low temperature solid-state synthesis was employed to dope LaB_6_ with europium (Eu), forming a single phase of La_x_Eu_1−x_B_6_ nanoparticles [71]. Combining divalent Eu^2+^ and trivalent La^3+^ changed the free electron density of the system, resulting in a shift of the plasmon when changing the ratios of the two metals (Figure 4D). Increasing the concentration of Eu^2+^ significantly increased the number of metal vacancies in the structure, which allowed the plasmon to be tuned across a very wide range (1100 nm to 2050 nm).

### 3.2. Nanoparticle Size 

In addition to doping, controlling the size of phase-pure LaB_6_ also offers a means of control when tuning the position of the plasmon, where shifts of size and shape change the particle surface enough to influence the resonant frequency [72]. However, due to past difficulties in synthesizing LaB_6_ nanoparticles directly, it has been difficult to directly synthesize a series of pure LaB_6_ with varying nanoparticle sizes. 

Theoretical models using DFT clearly show how particle sizes ranging from 10 nm to 100 nm can shift the LSPR, where the larger the particle the longer the wavelength absorbed so long as the particles are smaller than 80 nm (Figure 5A) [73]. In contrast, LaB_6_ nanoparticles larger than 80 nm are too big to influence the plasmon and have a lower efficiency for NIR absorbance. Experiments of particles below 5nm have shown that even a small difference in size has a notable impact on the position of the plasmon. For example, increasing the particle size of phase pure LaB_6_ from 2.1 nm to 4.7 nm red-shifts the absorbance from 1080 nm to 1250 nm, and the same red shift trend is observed when a diol-based ligand is included, where changing the size from 2.5 nm to 2.8 nm moves the plasmon from 1190 nm to 1220 nm (Figure 5B) [74]. Interestingly, with particles well below 10 nm in size the DFT calculations predicting the position of the plasmon are lower than the actual values, and further work is needed to understand what additional contributions from such small sized particles influence the LSPR theoretical calculations.

Calculations have also found that the scattering efficiency changes when adding different sizes of LaB_6_ to a polymer matrix (Figure 5C), which ought to be considered when developing films for window coatings. Decreasing the nanoparticle size from 200 nm to 50 nm significantly red-shifts the plasmon, and not only increases the intensity of the scattering but broadens the scatter as well. It should also be noted that the shape of the scattering coefficient curve changes in particles larger than 80 nm, which may be indicative of size-dependent LSPR behavior. When attempting to design a material that will capture a wide wavelength range, like the solar heat from 750 nm to 1200 nm mentioned above, it is ideal to use nanomaterials with a broad and intense absorbance peak [30].

Bulk sized LaB_6_ may also be reduced in size through grinding in the presence of a surfactant such as dodecylbenzene-sulfonic acid (DBS). The DBS makes it easier to disperse ~100 nm particles into ethylene glycol, with the absorbance intensity increasing in higher particle concentrations (Figure 5D) [75]. This makes LaB_6_ a feasible NIR photothermal conversion material. 

### 3.3. Morphology

In addition to size, the plasmonics of nanoparticles can also be tuned by altering the morphology of the system, where introducing different facets has a large impact on how free electrons behave on the surface [76]. Unfortunately, publications describing more than one nanoparticle shape of LaB_6_ are uncommon due to synthetic limitations, and those connecting shape and LSPR are even more rare. Whether morphological limitations are due to the chemistry itself or if low temperature methods simply need additional time to study has yet to be determined. The shape of LaB_6_ nanoparticles tends to be described as generic nanoparticles (assumed spheres) versus cubes [56,77] (Figure 6A) or as nanowires [78,79,80], but to our knowledge only two publications to date tie plasmonics to varying particle shape [81,82]. 

The optical response of LaB_6_ nanoparticles of varying sizes and shapes was compared using the discrete dipole approximation and experimental results were in agreement with the findings [82]. Comparing the extinction efficiencies of cubic and spherical particles of the same diameter (Figure 6B), it is clear that the optical properties are significantly influenced by the shape of the particles and that the nanocubes exhibit stronger NIR extinction. The Mie integration method is also an effective means of estimating optical properties of nanoparticles of changing sizes and shapes [83]. For LaB_6_, it was found that increasing the aspect ratio of spheroids can enhance the LSPR properties in the NIR [81]. In oblate spheroids, decreasing the aspect ratio has the effect of red-shifting the LSPR, and can significantly broaden the absorbance peak (Figure 6C).

## 4. Improving LaB_6_ Plasmonic Applications 

In order to make LaB_6_ nanoparticles feasible as coatings for various solar energy and window applications, researchers must gain an understanding of plasmonic behaviors when incorporating the particles into films. Once the surrounding media is changed, the absorbance and intensity of the LSPR signal is also altered. To make cleaner films, LaB_6_ nanoparticles can be put into solutions using various polymer matrixes [84,85]. As with the changing particle concentrations mentioned above, varying the size of LaB_6_ nanoparticles (while maintaining the same concentration) in polymethyl methacrylate (PMMA) composites changes the absorbance intensity (Figure 7A), and the ideal particle size to achieve maximum intensity is just below 100 nm [86]. A similar effect has been observed in the transmittance and reflectance profiles of LaB_6_ nanoparticles dispersed in acrylic coatings, where decreasing the LaB_6_ concentration increases the transmittance intensity (Figure 7B) [87].

Creating visibly clear films with LaB_6_ remains an ongoing challenge. Since LaB_6_ nanoparticles are typically made with no ligands, they tend to aggregate with no physical barrier keeping them apart. To improve upon this, researchers are starting to develop methods to incorporate surfactants directly on the surface of LaB_6_. For example, using cetyltrimethyl ammonium bromide (CTAB) as a surfactant coating makes it possible to stabilize and reduce the agglomeration of LaB_6_ nanoparticles in water (Figure 7C). There has also been a report of incorporating a ligand during the LaB_6_ reaction using a low temperature method to convert isophthalic acid to 1,3-phenylenedimethanol in-situ, with the diol binding to the LaB_6_ surface as soon as it is formed in-situ [74]. As a result, the ligand-bound LaB_6_ nanoparticles can be embedded directly into various polymer matrixes without the need for additional treatment, producing unaggregated and clear films composed of LaB_6_ in PMMA, polystyrene, or tetraethoxyorthosilane (TEOS) glass (Figure 7D). While this review focuses on tuning the plasmon of LaB_6_, it is also important to note that there are many more applications for this material beyond building specialized windows for heat control. Many new applications of LaB_6_ nanoparticles have come to light in recent years. For example, the photothermal conversion properties of LaB_6_ are being explored for biomedical applications like cancer [75,88,89,90]. LaB_6_ nanoparticles also interact with bacteria, and studies focusing on ablation and hydrogen production are ongoing [91,92]. Looking beyond nanoparticles to nanowires, LaB_6_ is now known to be an efficient field emitter [78,79,80]. There are many more potential applications, and this list offers you a small sampling of how this material is advancing multiple fields of research.

### Future Challenges of Plasmonic LaB_6_

Much has been learned about the optical behavior and tunability of LaB_6_, and work is ongoing. Perhaps the unspoken secret of this research space is how challenging it is to develop the synthetic procedures to make small phase-pure LaB_6_ nanoparticles. Ball milling bulk particles to reduce the particle size introduces contaminants and does not maintain uniform morphology. Cutting single crystals into nano-sized particles is difficult and does not translate easily to larger scale [78]; plasma methods do not offer a means of easy control for tuning [93], etc. 

To further complicate matters, the mechanism behind the formation of the B_6_ cluster is still unknown. For decades it was assumed that high temperatures (>1200 °C) and pressures were energetically required to make LaB_6_, but that is not the case [94,95]. It has also been assumed that a reducing agent such as magnesium was required to drive the reaction [59,96], but new reports find this to be untrue [74,95]. Only recently was it found that the halogen of the lanthanum salt influences the reaction to expand or contract the crystal lattice, and can act as a bridging ligand between La atoms in low temperature reactions [48,58].

In order advance this field it is becoming increasingly important to build a better understanding of the chemistry involved in the formation of LaB_6_. In learning more about the mechanism, it will be much easier to tune these particles for optical applications as well as electronics, thermionics, and solar energy. With a better understanding of the formation of the robust boron clusters in LaB_6_ and the potential interplay of boron and halogens, we will likely discover new applications previously unexplored.

## 5. Conclusions

LaB_6_ is an intriguing plasmonic material that has gained much attention in the last decade for its ability to intensely absorb in the NIR region of the electromagnetic spectrum. Large advances have been made in developing methods to finely tune the position of the plasmon, making the shift from the visible region to the NIR possible. This review explains how some of the fundamentals of the LaB_6_ framework influence the optical properties, and how the plasmon can be tuned by controlling lattice defects, particle size and shape, and through doping. As advances continue to be made in this field, it is hoped that a there will be a better understanding of how the structure forms, enabling better control over this system and potentially opening the door to new applications in need of stable plasmonic nanoparticles.

## Figures and Tables

**Figure 1 materials-11-02473-f001:**
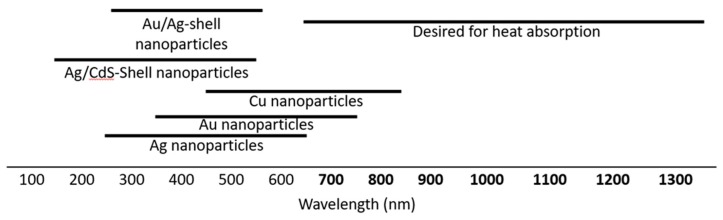
Typical absorbance range of plasmonic metal nanoparticles: Ag [15], Au [16], Cu [17], Ag/CdS-shell [18], Au/Ag-shell [19], and the desired range to absorb solar heat.

**Figure 2 materials-11-02473-f002:**
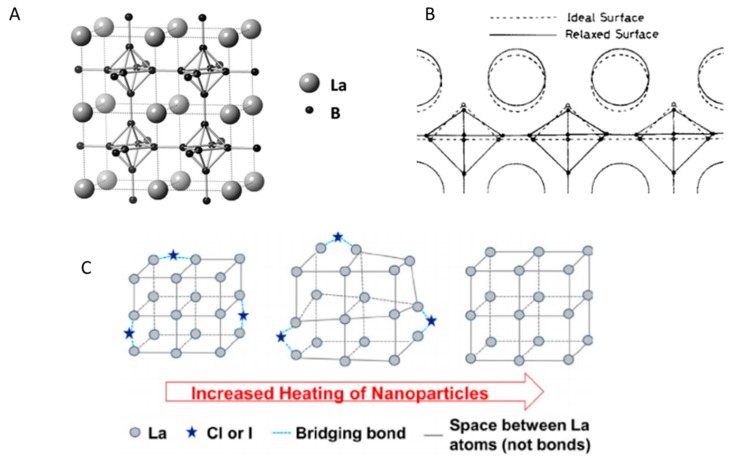
(**A**) Four unit cells of lanthanum hexaboride (LaB_6_) with a lattice constant of approximately 4.15 Å (reprinted from [48]). (**B**) Position of La (circle) and B_6_ (diamond) in relaxed and ideal states on LaB_6_ surface (reprinted from [49]). (**C**) Removal of halogen bridge between La atoms with continued heating of LaB_6_ nanoparticle (reprinted from [58]).

**Figure 3 materials-11-02473-f003:**
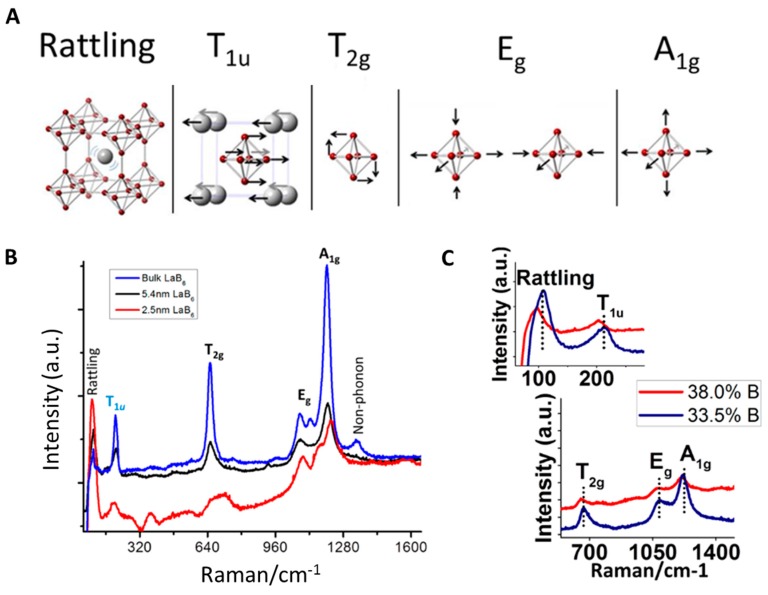
(**A**) Raman active vibrational modes of LaB_6_; (**B**) vibrational mode changes with changing nanoparticle sizes; and (**C**) shifting vibrational modes of 6.2 nm particles with varying B content (reprinted from [63]).

**Figure 4 materials-11-02473-f004:**
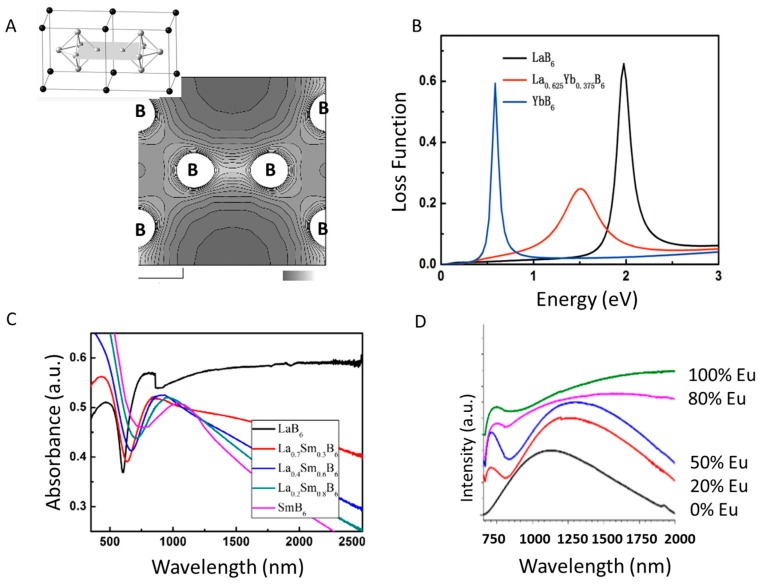
(**A**) Total charge density map of LaB_6_ (grey scale shows increasing charge density) (reprinted from [67]). (**B**) Energy loss function of LaB_6_, La_0.625_Yb_0.375_B_6_ and YbB_6_ in the low energy region (reprinted from [69]). (**C**) Absorbance spectra of SmB_6_, La_0.2_Sm_0.8_B_6_, La_0.4_Sm_0.6_B_6_, La_0.7_Sm_0.3_B_6_ and LaB_6_ (reprinted from [70]). (**D**) Absorbance spectra of La_x_Eu_1−x_B_6_ changing with Eu concentration (normalized) (reprinted from [71]).

**Figure 5 materials-11-02473-f005:**
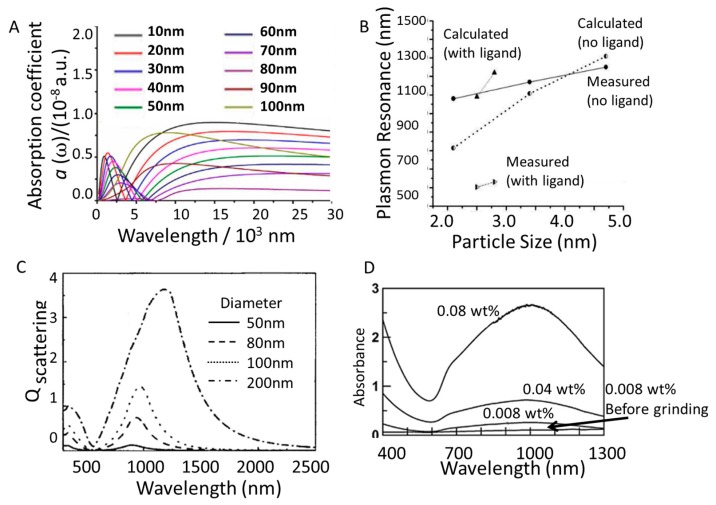
(**A**) Modeled absorption spectrum for LaB_6_ nanoparticles ranging from 10 nm to 100 nm (reprinted from [73]). (**B**) Calculated and measured (via diffuse reflectance) LSPR peak positions of ligand-free and ligand-bound LaB_6_ nanoparticles (reprinted from [74]). (**C**) Calculated scattering efficiency of single spherical LaB_6_ nanoparticles embedded in a polymer for sizes ranging from 50 nm to 200 nm (inset shows fraction of scattering to total extinction efficiency at 500 nm) (reprinted from [30]). (**D**) Absorption spectra for ethylene glycol dispersion of LaB_6_ powders before and after grinding at different concentrations (reprinted from [75]).

**Figure 6 materials-11-02473-f006:**
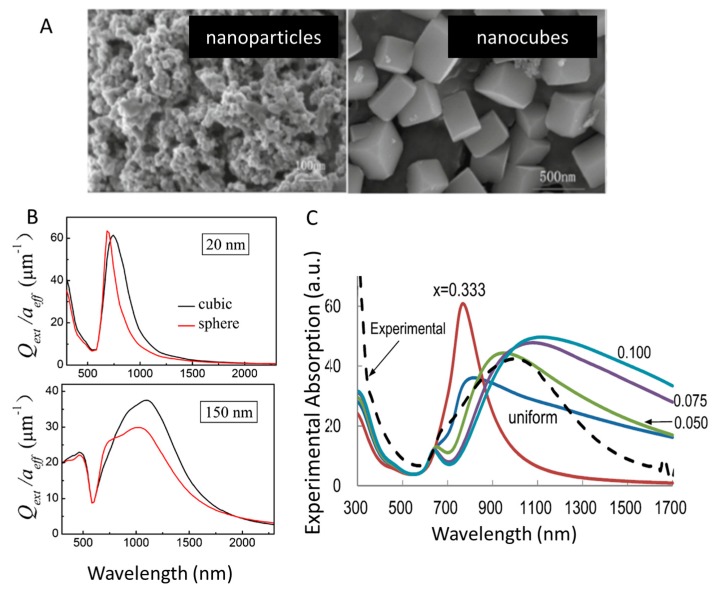
(**A**) SEM images of LaB_6_ nanoparticles and nanocubes (reprinted from [57]). (**B**) Comparison of extinction efficiencies divided by effective radius between cubic and spherical LaB_6_ particles (reprinted from [82]). (**C**) Absorption cross sections calculated for 100,000 LaB_6_ oblate particles with different aspect ratios and a standard deviation of 0.1 (reprinted from [81]).

**Figure 7 materials-11-02473-f007:**
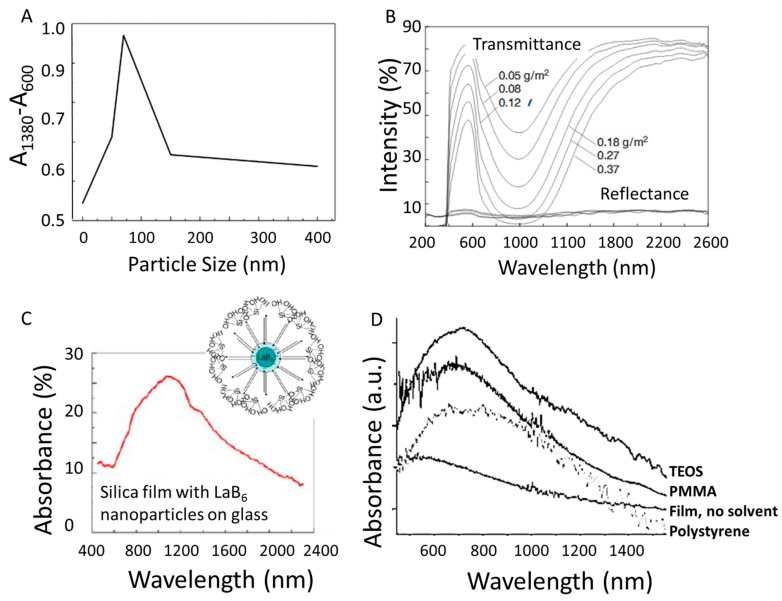
(**A**) Relative strength of absorption in composites containing LaB_6_ particles of different sizes; 0 nm represents pure polymer without LaB_6_. (reprinted from [86]). (**B**) Transmittance and reflectance profiles of LaB_6_ nanoparticle-dispersed acrylic coatings with different thicknesses on PET films (reprinted from [87]). (**C**) Absorbance spectra of cetyltrimethyl ammonium bromide (CTAB)-stabilized LaB_6_ nanoparticles (reprinted from [84]). (**D**) Absorbance spectra of 2.8nm ligand-bound LaB_6_ nanoparticles embedded in polymethyl methacrylate (PMMA), polystyrene, and tetraethoxyorthosilane (TEOS)-derived glass (reprinted from [74]).

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
