# Peer review of "Tuning the Surface Plasmon Resonance of Lanthanum Hexaboride to Absorb Solar Heat: A Review"

_materials, 2018, doi:10.3390/ma11122473_

Round 1
Reviewer 1 Report
This article is concerning with a plasmonic behavior of LaB6, an intermetallic compound. From the viewpoints of light absorption/reflection applications, plasmonic behaviors of LaB6 nanoparticles are reviewed. It can be summarized as below:
- Fundamental description of LaB6 nanoparticles: Deformation of crystal lattice on the surface of nanoparticle changes the plasmonic absorption band.
- Doping the other lanthanides, Y, Sm, an Eu, reduces the absorption and causes a red-shift of plasmon band of LaB6.
- Size dependency: The larger nanoparticles result in the higher (longer) wavelength of plasmon resonance.
- Morphological effect: Cubic particles are better than the spherical particles for NIR absorption.
- Dispersion of LaB6 for applications.
In my opinion, this article is very interesting. The plasmonic behavior of LaB6 is not well-known, and the structural control is a challenge in the material science. Therefore, this article will encourage the researchers concerning with the plasmonic materials.
However, I found some points to be considered before publication in Materials. I listed them below. I wish my comments will help to improve the manuscript.
(1) Introduction
- Line 28: “leaving the near infrared (NIR) region inaccessible”. However, the Au nanowire and nano-shell can have the LSPR band in the NIR region.
• J. Phys. Chem. B 2006, 110, 7238-7248
• J. Am. Chem. Soc., 2014, 136 (24), pp 8489–8491
These Au nanomaterials are applied for the biomedical applications (imaging, hyperthermia, and diagnosis). Therefore, this statement in the introduction should be changed. It can be associated to the other reasons, such as mechanical/thermodynamic instability of Au nanostructures.
(2) Explanation
- In figure 2A, the lattice constants of the LaB6 should be added for easy understanding in terms of size dependency.
- Line 193-194: “the larger the particle the longer wavelength”. However, the figure 5A suggests the exceptional behavior for the largest particles (90 and 100 nm). In the original article (ref 71), it is mentioned that “While the sizes are larger than 80 nm, the LSPR resonance peaks disappear” (page 1098). That is, the LaB6 nanoparticles larger than 80 nm can’t localize the surface plasmon. This point can be associated to the other part: In the figure 5C, the curve of scattering coefficient implies the change around 80 nm. In the figure 7A, the larger particles (> 100 nm) have low efficiency for NIR absorption. This can be important to understand the size dependency of this material.
(3) Missing description
- Figure 4B, the meaning of gray scale is required (high/low density etc.).
- Figure 4B, the Y axis “Loss function” or something is required.
- Figure 4, caption for C and D
(4) Typos
I found many typos. Please check the manuscript again.
(“growth. .” in Line 67, “LaB6.In” in 70, “heated the” in 83, “temperature The” in138, “itself or or if” in 226, etc.)
Author Response
Dear Reviewer 1,
Thank you for taking the time to review our manuscript. We appreciate the constructive feedback that we received and have done our best to respond to your concerns, which we believe have helped to strengthen this Review article. Please note that the additions to the manuscript made in response to both Reviewers’ comments are highlighted in yellow in the attached revision.
If you have any additional concerns please do not hesitate to let us know.
RESPONSE TO REVIEWER 1 COMMENTS:
In my opinion, this article is very interesting. The plasmonic behavior of LaB6 is not well-known, and the structural control is a challenge in the material science. Therefore, this article will encourage the researchers concerning with the plasmonic materials.
However, I found some points to be considered before publication in Materials. I listed them below. I wish my comments will help to improve the manuscript.
1. - Line 28: “leaving the near infrared (NIR) region inaccessible”. However, the Au nanowire and nano-shell can have the LSPR band in the NIR region.
• J. Phys. Chem. B 2006, 110, 7238-7248
• J. Am. Chem. Soc., 2014, 136 (24), pp 8489–8491
These Au nanomaterials are applied for the biomedical applications (imaging, hyperthermia, and diagnosis). Therefore, this statement in the introduction should be changed. It can be associated to the other reasons, such as mechanical/thermodynamic instability of Au nanostructures.
We thank the Reviewer for bringing this to our attention. We had neglected to look at the properties of nanowires in the introduction and have corrected this in the manuscript. The updated phrasing in Line 28 of the intro now says the following: “in traditional plasmonic metals (Figure 1). This leaves the near infrared (NIR) region mostly inaccessible to metal nanoparticles, with the exception of novel engineered geometries of some metals, such as the cases of Au nanowires and shells.”
2. In figure 2A, the lattice constants of the LaB6 should be added for easy understanding in terms of size dependency.
The description of Figure 2A now says the following: “Four unit cells of LaB6 with a lattice constant of approximately 4.15Å”
Line 193-194: “the larger the particle the longer wavelength”. However, the figure 5A suggests the exceptional behavior for the largest particles (90 and 100 nm). In the original article (ref 71), it is mentioned that “While the sizes are larger than 80 nm, the LSPR resonance peaks disappear” (page 1098). That is, the LaB6 nanoparticles larger than 80 nm can’t localize the surface plasmon. This point can be associated to the other part: In the figure 5C, the curve of scattering coefficient implies the change around 80 nm. In the figure 7A, the larger particles (> 100 nm) have low efficiency for NIR absorption. This can be important to understand the size dependency of this material.
We thank the reviewer for pointing out this error, and agree that there should be a statement in the manuscript to highlight the importance of the size range where LSPR can be tuned in LaB6. We have updated the text discussion Figure 5A to include the following: “…where the larger the particle the longer the wavelength absorbed so long as the particles are smaller than 80 nm (Figure 5A) [73]. In contrast, LaB6 nanoparticles larger than 80 nm are too big to influence the plasmon and have a lower efficiency for NIR absorbance.”
We have also added additional text to mention the size-dependent behavior of Figure 5C by including the following text: “It should also be noted that the shape of the scattering coefficient curve changes in particles larger than 80 nm, which may be indicative of size-dependent LSPR behavior.”
3. Missing description
Figure 4B, the meaning of gray scale is required (high/low density etc.).
Figure 4B, the Y axis “Loss function” or something is required.
Figure 4, caption for C and D
Figure 4 has been updated to include the y axis in Figure 4B. The caption for Figure 4 has also been updated to incorporate the suggested corrections. The description now states the following: “Figure 4. A) Total charge density map of LaB6 (grey scale shows increasing charge density). B) Energy loss function of LaB6, La0.625Yb0.375B6 and YbB6 in the low energy region. C) Absorbance spectra of SmB6, La0.2Sm0.8B6, La0.4Sm0.6B6, La0.7Sm0.3B6 and LaB6. D) Absorbance spectra of LaxEu1−xB6 changing with Eu concentration (normalized).”
(4) Typos
I found many typos. Please check the manuscript again.
(“growth. .” in Line 67, “LaB6.In” in 70, “heated the” in 83, “temperature The” in138, “itself or or if” in 226, etc.)
We thank the Reviewer for pointing out the typos to us, and have corrected the typos throughout the manuscript in addition to the above noted.

Reviewer 2 Report
If one reads this manuscript, one can easily notice that the overall English writing has no issues. However, as a scientist, I am not here to read the English wiring style but to review. The writing of this manuscript is the worst. I highlight many places where I had significant trouble understanding the scientific meaning of their writing. Below I list my suggestions.
Page 1 Line 2: The title is vague. Plasmons are quasiparticles free electrons and It's not clear to me from this title what is being tuned here. If it is plasmonic nanoparticle, authors should mention this.
Page 1 Line 12: Window applications. You are reading a paper on optical material and all of suddenly from nowhere it mentions “window applications” Is this the same window that is used in the residential homes, public restrooms, park shades, airplane window, and so on, one can go on naming it. If the aim is to replace the ordinary glass windows available in the market by the improved windows that can effectively control the flow of heat getting in and out of the rooms during both the hot and cold temperature, this should be clarified.
Page 1 Line 13: Do the authors mean nanoparticle by the word “structure” here? If so, this must be clearly stated. “structure” is a general term. These authors need to be more specific to what they are referring too.
Page 1 Line 14: Do the authors mean energy and absorption properties by the word “these optical properties”? Please elaborate on it.
Page 1 Line 15: Plasmon position. What do the authors mean by plasmon position here? Do they mean a position where the resonance takes place? Also, what are these uncover properties mean here? Physical properties, optical properties, chemical properties, microstructural properties? Authors may find the following definition useful:
“Observe, measure and communicate: Scientists use their senses when solving science problems. They use their eyes to spot details. They use their noses to detect if something is stinky. They use their hearing, touch, and even sense of taste.” (Random source: online). Effective communication is what I sense is missing in this abstract.
I encourage authors to critically and carefully choose the words and re-write the abstract considering the general audiences.
Page 1 Line 20: Noble metals such as Ag, Au, and Cu nanoparticles are well known for their plasmonic properties, they typically only absorb in the ultraviolet and visible regions.
Page 1 Line 24: What do the author mean by “their sensitivity to small changes”? here. The sentence is not clear. What do the authors mean by “sensitivity” here?
Page 1 Line 26: Metals, nanoparticles and here authors refer to metamaterial without giving an example of metamaterial or without citations.
Page 1 Line 33-34: I encourage authors to add a bit more information about this figure. While the color print looks ok to me, if printed it in gray, I am afraid it would look like just fonts placed every where. For example, authors can use horizontal arrows to define the wavelength ranges for each core/shell nanoparticles as well as for desired ….,
Page 2 Line: Now it says “.. ideal for window coatings”. That means readers must go all the way to the second page to know the meaning of windows applications. Both the “Title and Abstracts are misleading to the readers. I encourage authors to revise it as most readers do not read beyond abstract unless the subject is related to their own researcher.
Page 2 Line 37: Is it true that Lanthanum hexaboride (LaB6) is a plasmonic metal? It is an inorganic chemical? According to Wikipedia, “A metal is a material that, when freshly prepared, polished, or fractured, shows a lustrous appearance, and conducts electricity and heat relatively well.”
Page 2 Line 39: LaB6 also absorbs light very strongly at about l (insert lambda here) = 1000nm.
Page 2 Line 44: Add “particles” following the word LaB6. Authors should compare apples with apples only and not with oranges.
Page 2 Line 47: This is good, this shows authors are familiar and know what the surround media would be like “surround media (i.e. ligands and polymer matrices).” Do the author mean “optical properties” by “properties” here? Please state it clearly.
Page 2 Line 58: LaB6 is now referred to as: LaB6 is composed of interconnecting hexaboride clusters with lanthanum (La) atoms residing in the interstitial spaces.
Fig 2 symbols are misleading. Label it correctly.
This paper is written in haste and consists many flaws in it. Consistency in the use of terminology is several lacking. I do not think I can go further reviewing from page 3 to 12 (its never ending). Without the authors sufficiently revising the manuscript, I can not recommend publication of this manuscript as is.
Author Response
Dear Reviewer 2,
We appreciate the feedback that we received and have done our best to respond to your concerns, which we believe have helped to strengthen this review article. Please note that the additions to the manuscript made in response to both Reviewers’ comments are highlighted in yellow in the attached revision.
If you have any additional concerns please do not hesitate to let us know. Thank you for your time.
RESPONSE TO REVIEWER 2 COMMENTS:
If one reads this manuscript, one can easily notice that the overall English writing has no issues. However, as a scientist, I am not here to read the English wiring style but to review. The writing of this manuscript is the worst. I highlight many places where I had significant trouble understanding the scientific meaning of their writing. Below I list my suggestions.
1. Page 1 Line 2: The title is vague. Plasmons are quasiparticles free electrons and It's not clear to me from this title what is being tuned here. If it is plasmonic nanoparticle, authors should mention this.
There seems to be some miscommunication here. Quasiparticles are not free electrons (if I understand the Reviewer’s comment correctly), however, we agree that the title could be more descriptive, and have changed the title to “Tuning the Surface Plasmon Resonance of Lanthanum Hexaboride to Absorb Solar Heat: A Review”
2. Page 1 Line 12: Window applications. You are reading a paper on optical material and all of suddenly from nowhere it mentions “window applications” Is this the same window that is used in the residential homes, public restrooms, park shades, airplane window, and so on, one can go on naming it. If the aim is to replace the ordinary glass windows available in the market by the improved windows that can effectively control the flow of heat getting in and out of the rooms during both the hot and cold temperature, this should be clarified.
We have deleted the text “for various window applications” from the abstract.
3. Page 1 Line 13: Do the authors mean nanoparticle by the word “structure” here? If so, this must be clearly stated. “structure” is a general term. These authors need to be more specific to what they are referring too.
Page 1 Line 14: Do the authors mean energy and absorption properties by the word “these optical properties”? Please elaborate on it.
Page 1 Line 15: Plasmon position. What do the authors mean by plasmon position here? Do they mean a position where the resonance takes place? Also, what are these uncover properties mean here? Physical properties, optical properties, chemical properties, microstructural properties? Authors may find the following definition useful: “Observe, measure and communicate: Scientists use their senses when solving science problems. They use their eyes to spot details. They use their noses to detect if something is stinky. They use their hearing, touch, and even sense of taste.” (Random source: online). Effective communication is what I sense is missing in this abstract.
I encourage authors to critically and carefully choose the words and re-write the abstract considering the general audiences.
We are unclear as to what the Reviewer was intending to convey with the “randomly sourced” internet quote, so we will leave that point out of our rejoinder and respond instead to the more scientifically germane points below. If we inadvertently missed a substantial point there, we are happy to reply.
It is with deliberate intent that we used the word “structure” in the manner we did. As the Reviewer is aware, this term is used throughout physics and chemistry variously in a way that depends on context – e.g. electronic structure, crystal structure, chemical structure, etc. This paper describes the fundamentals behind the crystallographic structure (i.e. position of atoms) of LaB6, explaining how vibrations and atomic positions shift with decreasing particle size.
We are genuinely sorry the Reviewer had difficulty with the abstract. In order to clarify all of the problematic points mentioned above, we have rewritten the second half of the abstract to better consider the general audiences.
The new text is reproduced here: “Given the growing popularity of LaB6, this review focuses on the advances made in the past decade with respect to controlling the plasmonic properties of LaB6 nanoparticles. This review discusses the fundamental structure of LaB6 and explains how decreasing the nanoparticle size changes the atomic vibrations on the surface and thus the plasmonic absorbance band. We explain how doping LaB6 nanoparticles with lanthanide metals (Y, Sm, and Eu) red-shifts the absorbance band and describe research focusing on the correlation between size dependent and morphological effects on the surface plasmon resonance. This work also describes successes that have been made in dispersing LaB6 nanoparticles for various optical applications, highlighting the most difficult challenges encountered in this field of study.”
4. Page 1 Line 24: What do the author mean by “their sensitivity to small changes”? here. The sentence is not clear. What do the authors mean by “sensitivity” here?
It is well known that the plasmon in LSPR materials are sensitive to small changes, meaning minor alterations to the structure or the particles surroundings may shift the plasmon, such as defects, vacancies, etc. To clarify this to readers unfamiliar with the sensitivity of the plasmon to such conditions, we have added the following text: “(e.g. size, morphology, atomic vacancies, etc.)”
5. Page 1 Line 26: Metals, nanoparticles and here authors refer to metamaterial without giving an example of metamaterial or without citations.
To simplify the text without additional explanation to confuse the reader, we have changed “plasmonic metamaterials” to “plasmonic metals.” This does not change the context of the text. We apologize for causing any confusion in changing the terminology.
6. Page 1 Line 33-34: I encourage authors to add a bit more information about this figure. While the color print looks ok to me, if printed it in gray, I am afraid it would look like just fonts placed every where. For example, authors can use horizontal arrows to define the wavelength ranges for each core/shell nanoparticles as well as for desired ….,
The Reviewer makes a useful point. Figure 1 has been redone so that it will be clear in both color and black & white print.
7. Page 2 Line: Now it says “.. ideal for window coatings”. That means readers must go all the way to the second page to know the meaning of windows applications. Both the “Title and Abstracts are misleading to the readers. I encourage authors to revise it as most readers do not read beyond abstract unless the subject is related to their own researcher.
Following the Reviewer’s suggestions, the Title has been altered and the bulk of the abstract rewritten. In the caption mentioned here, the text “ideal for window coatings” has been replaced with “desired range to absorb solar heat.”
8. Page 2 Line 37: Is it true that Lanthanum hexaboride (LaB6) is a plasmonic metal? It is an inorganic chemical? According to Wikipedia, “A metal is a material that, when freshly prepared, polished, or fractured, shows a lustrous appearance, and conducts electricity and heat relatively well.”
LaB6 is widely considered to be a plasmonic metal in the scientific community. While I agree that Wikipedia (or Google searches more broadly) are a possible resource for those who are unfamiliar with a topic, it is not a peer reviewed journal and offers only trivial depth. It is not a scientifically accepted resource (by AAAS, NPG, etc.). In that sense, there is no value in responding to a non-scientific objection if we are upholding scientific standards for publication. Phenomenological properties such as luster were used centuries ago as a heuristic before the physical sciences were well-developed. A more commonly accepted standard would be that of the temperature-dependence of the conductivity, which is commonly used by materials scientists, physicists, and engineers. The simplest case of inorganic materials being conductors would be the broad class of inorganic superconductors, which tend to exhibit metallic conductivity above their transition temperature. In the most fundamental sense, there are only insulators (to varying degrees) and metals. Given that the most common use of LaB6 is as an electron emitting cathode in electron microscopes, this matter should be clear.
9. Page 2 Line 39: LaB6 also absorbs light very strongly at about l (insert lambda here) = 1000nm.
1000nm is an implied wavelength, so no lambda is required
10. Page 2 Line 44: Add “particles” following the word LaB6. Authors should compare apples with apples only and not with oranges.
We added the word “nanoparticles” following LaB6 here.
11. Page 2 Line 47: This is good, this shows authors are familiar and know what the surround media would be like “surround media (i.e. ligands and polymer matrices).” Do the author mean “optical properties” by “properties” here? Please state it clearly.
We have added the word “optical.”
12. Page 2 Line 58: LaB6 is now referred to as: LaB6 is composed of interconnecting hexaboride clusters with lanthanum (La) atoms residing in the interstitial spaces.
This is a true statement. LaB6 is made of interconnecting layers of B6 clusters with La atoms in the interstitial (open) spaces throughout the network of B. We are unclear as to what the issue is here (if any).
13. Fig 2 symbols are misleading. Label it correctly.
We are happy to clarify things where we can, but it is unclear which symbols are misleading. The Reviewer should note that these images are taken directly from published work and are being used with the written permission of the responsible Journals. Changing the symbols would alter the permissions for use in this publication.
14. This paper is written in haste and consists many flaws in it. Consistency in the use of terminology is several lacking. I do not think I can go further reviewing from page 3 to 12 (its never ending). Without the authors sufficiently revising the manuscript, I can not recommend publication of this manuscript as is.
We are sorry the Reviewer feels that this work was written in haste; it appears the Reviewer did not read the entire article based upon this comment. We were very thoughtful of the organization of this work and did our best to include all of the publications written in the past decade focusing on controlling the plasmonic properties of LaB6 nanoparticles, which was a massive undertaking. Furthermore, the terminology used throughout the text is consistent with what is found in the literature. We’re uncertain as to the specific objection here.
The length of this paper is as expected and consistent with other publications as this is a review article and not intended to be a brief research paper, communication or letter. The expectation of this type of publication is to thoroughly cover the topic with relevant references, and limiting this work to only two pages would be insufficient to describe the progress made in this field in the last decade.
We appreciate constructive feedback to make a publication stronger and have done our best to professionally respond to the suggestions and comments made by the Reviewer.